# Speech and Nonspeech Parameters in the Clinical Assessment of Dysarthria: A Dimensional Analysis

**DOI:** 10.3390/brainsci13010113

**Published:** 2023-01-07

**Authors:** Wolfram Ziegler, Theresa Schölderle, Bettina Brendel, Verena Risch, Stefanie Felber, Katharina Ott, Georg Goldenberg, Mathias Vogel, Kai Bötzel, Lena Zettl, Stefan Lorenzl, Renée Lampe, Katrin Strecker, Matthis Synofzik, Tobias Lindig, Hermann Ackermann, Anja Staiger

**Affiliations:** 1Clinical Neuropsychology Research Group (EKN), Institute of Phonetics and Speech Processing, Ludwig-Maximilians-University, 80799 Munich, Germany; 2Clinic for Psychiatry and Psychotherapy, Neurophysiology & Interventional Neuropsychiatry, University of Tübingen, 72076 Tübingen, Germany; 3Department of Neurology, Klinikum Großhadern, Ludwig-Maximilians-University, 81377 Munich, Germany; 4Clinic for Neuropsychology, City Hospital Munich Bogenhausen, 81925 Munich, Germany; 5Medical Clinic and Outpatient Clinic IV, Ludwig-Maximilians-University, 81377 Munich, Germany; 6Clinic for Neurology, Hospital Agatharied, 83734 Hausham, Germany; 7School of Medicine, Klinikum Rechts der Isar, Orthopedic Department, Research Unit for Pediatric Neuroorthopedics and Cerebral Palsy of the Buhl-Strohmaier Foundation, Technical University of Munich, 81675 Munich, Germany; 8Department of Logopedics, Stiftung ICP Munich, Center for Cerebral Palsy, 81377 Munich, Germany; 9Department of Neurodegenerative Disease, Hertie-Institute for Clinical Brain Research, German Center for Neurodegenerative Diseases (DZNE), and Center for Neurology, University of Tübingen, 72076 Tübingen, Germany; 10Department of Diagnostic and Interventional Neuroradiology, University Hospital Tübingen, 72076 Tübingen, Germany; 11Department of General Neurology, Hertie-Institute for Clinical Brain Research, University of Tübingen, 72076 Tübingen, Germany

**Keywords:** dysarthria assessment, clinical diagnostics, speech disorders, nonspeech tasks, diadochokinesis, DDK, validation

## Abstract

Nonspeech (or paraspeech) parameters are widely used in clinical assessment of speech impairment in persons with dysarthria (PWD). Virtually every standard clinical instrument used in dysarthria diagnostics includes nonspeech parameters, often in considerable numbers. While theoretical considerations have challenged the validity of these measures as markers of speech impairment, only a few studies have directly examined their relationship to speech parameters on a broader scale. This study was designed to investigate how nonspeech parameters commonly used in clinical dysarthria assessment relate to speech characteristics of dysarthria in individuals with movement disorders. Maximum syllable repetition rates, accuracies, and rates of isolated and repetitive nonspeech oral–facial movements and maximum phonation times were compared with auditory–perceptual and acoustic speech parameters. Overall, 23 diagnostic parameters were assessed in a sample of 130 patients with movement disorders of six etiologies. Each variable was standardized for its distribution and for age and sex effects in 130 neurotypical speakers. Exploratory Graph Analysis (EGA) and Confirmatory Factor Analysis (CFA) were used to examine the factor structure underlying the diagnostic parameters. In the first analysis, we tested the hypothesis that nonspeech parameters combine with speech parameters within diagnostic dimensions representing domain–general motor control principles. In a second analysis, we tested the more specific hypotheses that diagnostic parameters split along effector (lip vs. tongue) or functional (speed vs. accuracy) rather than task boundaries. Our findings contradict the view that nonspeech parameters currently used in dysarthria diagnostics are congruent with diagnostic measures of speech characteristics in PWD.

## 1. Introduction

### 1.1. Components of Clinical Dysarthria Assessment

Dysarthria is a complex neurogenic motor speech disorder. It involves dysfunctions of the respiratory, laryngeal, and supra-laryngeal motor subsystems in variable combinations and thereby leads to varied patterns of respiratory, vocal, articulatory, and prosodic disturbances in speech production. As a result of these deficits, dysarthric speech may be less intelligible and less efficient than normal speech, may sound less “natural”, and may require more effort and attention from conversation partners. In their daily life, persons with dysarthria (henceforth PWD) can therefore be challenged at all levels of social participation, that is, with partners, their family and friends, at work and in leisure, and in all requirements of independent living [1]. A thorough clinical evaluation of dysarthria must consider all dimensions of this health problem, which makes it an exceptionally complex and multifaceted endeavor.

The centerpiece of standard clinical dysarthria diagnostics by speech–language therapists (SLTs) is the assessment of speech characteristics directly related to the dysfunctions of the motor subsystems involved in speaking, as well as the prosodic consequences of these dysfunctions. The gold standard of this approach is based on auditory–perceptual speech parameters (e.g., [2,3,4,5,6,7]), but acoustic parameters are also becoming increasingly important as a clinical standard [4,8]. As a signature of impaired speech motor functions, auditory–perceptual and acoustic speech parameters are essential for speech–language therapists (SLTs) to draw conclusions about the pathomechanisms and subsystem involvements underlying a patient’s speech impairment and to design appropriate therapeutic interventions.

Another key domain of clinical dysarthria diagnostics is the assessment of communication-relevant speech parameters such as intelligibility or speech naturalness [9,10]. Describing how patients are perceived by their interlocutors, communication-relevant parameters do not distinguish between dysarthria types, pathomechanisms, or the involved motor subsystems and point more to functional outcome goals than to specific therapeutic strategies (e.g., [11,12]).

Along with the speech parameters mentioned so far, diagnostic parameters derived from *nonspeech* motor actions involving the respiratory, laryngeal, or supralaryngeal muscles have always played an important role in the clinical assessment of dysarthria [13]. They often make a substantial contribution to the diagnostic parameters used [3,8] or even replace speech altogether [14]. Nonspeech parameters are typically assessed through tasks requiring patients to sustain phonation as long as possible, repeat syllables or orofacial movements maximally fast, apply pressure to a tongue or lip sensor, or produce isolated nonspeech oral movements upon verbal command or imitation. For convenience, these parameters will henceforth be collectively referred to as “nonspeech”, though we are aware that some authors have used the terms “quasispeech” or “paraspeech” to classify tasks requiring maximally fast syllable repetitions or maximally long sustained vowel production (for taxonomies see [13]). However, though we use “nonspeech” as an umbrella term, the different subtypes of tasks that do not involve speaking will be kept separate throughout the article.

### 1.2. Clinical and Theoretical Issues in Assessing Speech Impairment through Nonspeech Tasks

#### 1.2.1. Effectors, Domain–General Functions, or Tasks?

Supplementing or substituting diagnostic speech parameters with nonspeech measures relies on the belief that they yield valid information about speech production in PWD, either as proxies for the speech disorder as a whole or for specific aspects of it. Particular value is seen in their effector-specificity, that is, their potential to assess the motor function of each articulator separately, which is perceived as an advantage because in speech, all subsystems are always recruited simultaneously, and the movements of the different articulators occur in such a confusingly rapid sequence that the detection of effector-specific impairments is challenging [15]. Therefore, maximum syllable repetition tasks (henceforth termed syllable diadochokinesis, DDK_syl_) use repetitions of /pa/, /ta/, and /ka/ with the aim of revealing the contributions of the lip, tongue blade, and tongue back functions, respectively, to the speech impairments of individuals with dysarthria [4]. Other oral–facial motor tasks are specifically designed to target isolated lip (e.g., lip smack, kiss), tongue (e.g., “make a click sound”, “stick out tongue”), or respiratory-laryngeal actions (e.g., “hum”, “clear throat”) to examine the contributions of the different articulators to the speech impairment in as pure isolation as possible. Diagnostic reports and test profiles of standard dysarthria assessments often merge these parameters with parameters derived from speech tasks, implicitly postulating that they measure the same thing. In the *Frenchay Dysarthria Assessment* (FDA-2; [3]) or the *Radboud Dysarthria Assessment* [8], for example, the contributions of articulatory, respiratory, and vocal impairments to dysarthria are conflated across different speech and nonspeech examinations, without distinguishing how possible dissociations between them would inform clinical decision making.

This practice builds on the tacit assumption that observations made in speech and nonspeech tasks have congruent clinical implications because they relate to the same effector organs. Theoretically, it relies on the idea that the motor pathologies underlying the dysarthrias have an effector-specific impact that transcends domain boundaries, and therefore the particularities of the overlearned and highly integrated organization of speech motor patterns are clinically negligible.

Another advantage of nonspeech approaches to speech assessment was seen in their potential to design tasks that have “goals that either mimic, or at least inform one about, the motor processes used in speech production” [15]; p. 140. In this vein, an integrative model of motor speech control proposed by Ballard and coworkers [16] described speech motor abilities as a family of domain–general motor functions, each of which can be tested by appropriately designed nonspeech tasks. Although the functional components that define speech according to this theory can hardly be fully specified, Ballard et al., cite timing or coordination as examples of functions that characterize skilled motor actions in all domains. As a nonspeech paradigm to study coordination in speech motor function, visuomotor tracking was widely discussed in experimental studies of dysarthria and apraxia of speech [16], but there is no compelling empirical evidence as to what exactly visuomotor tracking data tell us about a patient’s motor speech problems. Transferred to standard dysarthria assessment and the nonspeech parameters commonly used in clinical settings, the question arises as to which functional component of speech–motor impairment each parameter predicts (see Section 1.3).

A contrasting approach emphasizes that, while functional components such as timing or coordination are undoubtedly involved in any skilled voluntary motor action, they are orchestrated in specific ways depending on the task to be performed. Speech production is based on motor control processes that are extensively overlearned and must therefore be significantly shaped by experience-based neural plasticity mechanisms [17,18,19,20,21]. As an example of how motor patterns of the speech organs are determined by task demands, a study conducted by Westbury and Dembovski [19] revealed highly specific articulatory kinematics of maximally fast syllable repetition as opposed to speaking: Contrary to common assumptions, the healthy adult subjects included in this study slowed down rather than sped up their articulator movements, with the goal of achieving a maximally high repetition rate at substantially reduced movement amplitudes and short acceleration and deceleration times. This result suggests that high proficiency in syllable DDK requires the ability to switch from a speech mode to a motor control mode that is maximally efficient in producing as many PAs or TAs or PATAKAs as possible on a single breath, a goal that is fundamentally different from the motor goals of spoken language production. Such data, together with theoretical considerations based on dynamical system concepts of speech (e.g., [22,23]), have sparked a controversial and still ongoing discussion of functional organization principles of speech motor control and their clinical implications (e.g., [13,16,24,25,26,27,28,29,30,31,32,33,34,35]).

#### 1.2.2. “Pure” Motor Control

Another major avantage of nonspeech tasks is claimed to be that they separate motor from linguistic or psycholinguistic processes, opening a way to uncover the purely motor deficit underlying a speech disorder without “psycholinguistic goals getting in the way” [15]; p. 141. One of the motivations for strictly separating a linguistic and a motor level of speech production was clinical, that is, to be able to distinguish between “phonetic–motoric and linguistic impairments” [36], that is, between dysarthria or apraxia of speech on the one hand and aphasic–phonological impairment on the other.

The language-independence argument tacitly implies that skilled motor actions in general can be decoupled from their underlying cognitive goals, and nonspeech orofacial motor activities in particular are exempt from any other cognitive demands that might “get in the way” and affect their outcome. However, there is no empirical evidence so far that supports this assumption. If not linguistic, then other cognitive or motivational factors may come into play when individuals are instructed to move their speech organs in specific and unfamiliar ways. Nonspeech oral and facial motor tasks often require elaborate spatial awareness of the oral cavity and speech organs or the ability to implement complex movement instructions or imitate facial movements, and some require particular motivation and drive to achieve peak performance. These are all prerequisites individuals with brain damage do not always meet, and there are reports that such factors may influence performance or even cause null responses in both neurotypical and clinical populations [32,37,38].

#### 1.2.3. Face Validity

An important reason to call for a comprehensive validation of nonspeech parameters as measures of speech impairment is their lack of clinical face validity. Nonspeech parameters are not a priori linked to speech and communication. Low performance on nonspeech tasks is not a symptom that patients would ever complain about on their own. In fact, they usually don’t even notice problems of maximum syllable repetition rate or sustained phonation until they are confronted with such tasks in their first clinical examination and realize their inferiority to the SLT model. Even then, their primary desire is not to reduce these deficits, but to overcome their voice and articulation problems and regain intelligible and natural speech. Therefore, like any other clinical marker, the diagnosis of nonspeech oral–facial motor problems requires rigorous empirical validation to establish its clinical relevance as a measure of speech impairment. Note that this argument does not preclude nonspeech parameters from being useful as biomarkers, for example, in the early detection of a bulbar motor problem or in monitoring the progression of such a problem, which requires separate but equally rigorous validation [13,39].

Given these clinical considerations and the existing theoretical controversy, unproven a priori assumptions about the validity of nonspeech parameters as indicators of dysarthric speech are not warranted. As Folkins and coworkers wrote in their early contribution to this controversy, “it is not known, a priori, that nonspeech tasks will provide valuable information [for assessment and treatment], but it is worthwhile to ask the question—to see if they will” [15]; p. 139. With this in mind, empirical data are needed to answer this question and validate the roles of nonspeech tasks in clinical dysarthria diagnostics.

### 1.3. Empirical Relationships between Speech and Nonspeech Diagnostic Parameters

Despite the frequent use of nonspeech tasks in dysarthria assessment, only little is known to date about such relationships. Correlations between speech and nonspeech impairments are arguably abundant, as is generally the case in behavioral neurology and often due to third-variable effects [24]. However, hypothesis-guided investigations of specific speech–nonspeech relationships across relevant etiologies and dysarthria types are rare, and double dissociations indicating modality-specific speech-motor organization have often been reported [26,40].

Maximally fast syllable repetition (DDK_syl_), probably the most frequently used class of nonspeech tasks, has been considered an index of a variety of skills such as articulatory speed, motor coordination, articulatory motility, speech sequencing, lung capacity, or overall speech impairment (for a review see [39]), but there is no conclusive empirical evidence on exactly what speech problems DDK_syl_ parameters predict. At first sight, one might hypothesize that the maximum syllable repetition rate provides information about an individual’s articulation rate. On closer reflection, however, the assumption that a person’s ability to repeat a syllable at maximum speed provides information about their habitual speaking rate is not conclusive; after all, the habitual walking pace of a 100 m sprint champion is not necessarily different from that of a less athletic person, but their peak performance in a sprint competition certainly is. Accordingly, Neel and Palmer [41] found low and non-significant correlations between syllable DDK rates and reading rates in neurotypical individuals. Thus, opposing findings that dysarthric speakers show significant correlations between articulation rate and syllable DDK rate [31,42,43,44] apparently say more about the inability of patients to implement optimal strategies for DDK than about their articulation rate (cf. [45]).

Similarly, acoustic measures describing DDK_syl_ performance were proposed as a proxy for overall perceived speech impairment in patients with Parkinson’s disease [46] or more generally in dysarthria [14]. However, the available evidence is inconsistent (cf. [47]), and the predictive value of syllable DDK measures for dysarthria severity was found to vary substantially between etiologies, with an overestimation of severity in patients with cerebellar ataxia and an underestimation in progressive supranuclear palsy (PSP), and with a generally poor fit of regression models in cerebral palsy [33]. After all, it would be extremely surprising if a parameter that exclusively describes rapid repetitions of single syllables may capture, uniformly across all etiologies and dysarthria syndromes, the entire profile of a speech disorder as complex as dysarthria.

Similar problems exist for other types of nonspeech parameters. For example, the maximum vowel duration achieved in vowel prolongation tasks (henceforth referred to as *maximum phonation time*, MPT) was used as an index of pneumo-phonatory function in speech [3,4,8,48], but systematic correlation or regression analyses of MPT with speech breathing or voice functions in speech are rare. Based on their finding of a weak association of MPT with vital capacity, Solomon et al. [49] questioned the utility of sustained vowel production tasks for speech breathing assessment. Another longstanding controversy centers on the clinical significance of tongue or lip force measurements in predicting relevant features of dysarthric speech (e.g., [41,47,50]). Neil and Palmer [41] measured tongue strength in 57 healthy adults and found significant correlations with DDK performance, but not with speech rate. In a recent comprehensive study by Solomon and colleagues [51], only tenuous links between orofacial strength and perceptual measures of dysarthria were found, and even speakers with extreme weakness attained high intelligibility scores.

To summarize, there is only scant and inconsistent evidence so far about the links between nonspeech parameters used in clinical diagnostics and their possible correlates among the dimensions of dysarthric speech impairment. The lack of face validity of oral nonspeech parameters as indicators of speech impairment, the unresolved theoretical issues of a function- or effector-based organization of motor speech, and the dearth of consistent empirical data on the relationships between speech and nonspeech parameters call for further validation of the diagnostic parameters commonly used in clinical practice.

### 1.4. Study Design and Objectives

This is the first planned study to comprehensively address the validity of nonspeech parameters in the clinical assessment of speech impairment in patients with suspected dysarthria. The overarching objective of the study was to validate nonspeech parameters commonly used in dysarthria diagnostics by elucidating their relationships with a range of speech parameters in a large sample of patients with neurologic movement disorders. An equally large sample of control participants allowed us to standardize all parameters for their variability and for age and sex effects in a neurotypical population.

As a contribution to translational research in motor speech disorders, our study focused on diagnostic parameters that actually play a role in current clinical practice. Therefore, visuomotor tracking proficiency and kinematic or dynamic measures like amplitude, velocity, or maximum force of the articulators were not considered because they are not part of dysarthria assessment protocols used in standard clinical care. Furthermore, medical procedures such as electromyography or laryngoscopy were not included because they do not necessarily fall within the scope of SLTs.

The first goal of this study was to determine the structural links between the included diagnostic parameters and identify their latent underlying structure in order to understand the role of nonspeech parameters within a multivariate diagnostic approach. Exploratory graph analysis (EGA) followed by confirmatory factor analysis (CFA) [52,53] were used to determine the dimensionality of a set of 23 assessment parameters and how they combine into statistically distinct diagnostic factors. The research question was if speech and nonspeech parameters are conflated within dimensions, that is, share common traits, or if they are separated between dimensions, that is, represent distinct underlying traits.

The second objective was to test the hypothesis of task-specificity against two alternative hypotheses, that is, (a) whether speech and nonspeech diagnostic parameters combine according to effector-specific criteria, as suggested by most clinical applications of nonspeech measures and hypothesized in [15], and (b) whether they combine according to general functional goals, as hypothesized by Ballard et al. in their integrative model of oral–facial motor control [16]. To this end, a selection of speech and nonspeech parameters contrasting labial vs. lingual articulators as effectors and speed vs. accuracy as domain-general functional goals was subjected to further exploratory and confirmatory factor analyses.

Some of the questions raised here have partly been addressed in earlier studies of the same sample or different sub-samples [32,33,40,45,54], though with different methods, different research questions, and on a much smaller and less representative scale as far as the involved diagnostic parameters are concerned.

## 2. Materials and Methods

### 2.1. Participants

The study included 130 adult patients (age ≥ 18 years) with confirmed neurological diagnoses, that is, cerebrovascular accident (CVA), cerebral palsy (CP), cerebellar ataxia (ATX), Parkinson’s disease (PD), progressive supranuclear palsy (PSP), and primary dystonia (DYST). The main inclusion criterion was the presence of a motor impairment associated with the neurologic condition. This criterion was considered to be fulfilled without further inquiry in the etiologies classified as movement disorders, that is, PD, PSP, ATX, CP, and DYST. Patients after a stroke (CVA) were included only if they had clinical signs of a motor problem of the upper or lower extremities or the bulbar musculature. In order to be able to detect possible double dissociations of speech and nonspeech performance, the presence of dysarthria was not an inclusion criterion (cf. [40]). Exclusion criteria were (a) known non-neurologic diseases affecting the oral–pharyngeal–laryngeal apparatus or the respiratory system, (b) the presence of aphasia or apraxia of speech, and (c) the presence of additional neurologic or psychiatric disorders, including dementia (Mini-Mental-State-Examination (MMSE) < 20; [55]). All enrolled patients fully understood the instructions and had at least residual voice and speech to accomplish the experimental tasks. Table 1 provides an overview of demographic data.

The CVA subgroup (*n* = 26) included persons with ischemic (*n* = 18) or hemorrhagic strokes (*n* = 8) affecting frontal and temporal cortices, the corticobulbar tracts, basal ganglia, thalamus, cerebellum, and brain stem (10 left, 8 right, 8 bilateral). Twelve patients had multiple infarcts. All patients were diagnosed based on their medical history, neurologic examination, and structural brain imaging (CCT or MRT). Paresis of the upper and/or lower limb was present in 16 cases, and ataxia in 7. Seven patients had involvement of at least one of the cranial nerves V, VII, IX, X, XI, or XII. The patients were referred by the Department of Neuropsychology, City Hospital Bogenhausen, Munich.

The CP subgroup (*n* = 22) comprised adult persons with cerebral palsy (motor subtypes: 17 spastic, 3 dyskinetic, 2 spastic-dyskinetic). *Gross Motor Function Classification System* scores (GMFCS [56]) ranged across the five levels of the scale, with 16/22 cases having severe forms of cerebral palsy (GFCS IV, V). This subgroup was described in detail in [32]. The patients were referred by the Center for Cerebral Palsy, Munich.

The ATX subgroup (*n* = 24) included 22 participants with sporadic ataxias and 2 participants with spinocerebellar ataxias (one SCA 1, one SCA 6). SARA-Scores [57] ranged between 4 and 26 (median 14) on a scale from 0 to 40 (0 = no ataxia). Diagnoses were made based on medical histories, physical, and neurologic examinations, or were genetically confirmed. The patients were in- and outpatients of the Department of Neurology, University of Tübingen.

The PD subgroup (*n* = 25) comprised persons with idiopathic Parkinson’s disease (12 akinetic-rigid-dominant, 6 tremor-dominant, 7 equivalent subtypes). Hoehn and Yahr staging [58] was between 1 and 4 (median 2.5). The patients were recruited by the Departments of Neurology of the Ludwig-Maximilians-University Munich and the University Tübingen.

The PSP subgroup (*n* = 17) included 12 patients with PSP-Richardson’s Syndrome, 2 patients with PSP-Parkinsonism, and 3 patients with PSP-Corticobasal Syndrome. PSPRS-Scores [59] were between 11 and 48 (median 34). The patients were referred by the Department of Neurology, University of Munich.

The DYST subgroup (*n* = 16) included patients with primary dystonia and bilateral deep brain stimulation (DBS) of the globus pallidus internus (GPi). All patients had torticollis. In five patients, dystonia additionally affected other body parts (3 writer’s cramp, 2 Meige-syndrome, 1 spasmodic dysphonia). The patients were referred by the Department of Neurology, University of Munich. For DBS details see [54].

In all cases, participants continued their usual medication during the assessment. Patients on dopaminergic medication performed the tasks on-state. All patients with dystonia were examined in DBS-off-condition.

A control group of 130 healthy subjects also participated in the study (67 women, age evenly distributed between 18 and 90 years with a median age of 49; see Table 1, Figure A1 in Appendix C). Exclusion criteria were a diagnosis of neurological disease, or known diseases involving the oral–pharyngeal–laryngeal apparatus or the respiratory system.

All participants were native, monolingual speakers of German with normal or corrected-to-normal vision and age-appropriate hearing according to self-report. All had complete dentition or sufficient full or partial dental prostheses.

### 2.2. Assessments and Evaluation

Assessments took place in the referring clinic or in the participants’ homes. They were conducted by five certified speech–language pathologists (authors T.S., B.B., V.R., S.F, A.S.), all of whom were highly familiar with assessment procedures and auditory and acoustic speech analyses. The examination time was approximately 45 min.

The five examiners also performed the auditory and acoustic evaluations of speech and nonspeech parameters. Each examiner evaluated the samples of between 32 and 61 participants. All analysis steps were standardized prior to the evaluations, and all examiners completed training as described below.

The materials used for the assessment of speech parameters consisted of a short text (Appendix B, Table A3), two lists of disyllabic words and CV-syllables (Table A4 and Table A5), and sentence lists developed for intelligibility testing (see below). Text-, sentence-, and word production in all speech tasks was elicited in an audio-supported reading format, in order to alleviate the working memory demands of a pure repetition task and support reading in patients with reading problems. Each stimulus was first presented orthographically on the computer screen, together with a corresponding audio. Next, only the written text was presented and the participant was asked to read the stimulus aloud. No particular instructions regarding speaking rate were given. The auditory model was a 52-year-old male speaker with standard German pronunciation and an average articulation rate of 4.9 syllables per second. Speech samples were recorded using a directional condenser microphone (Røde NTG-1) and a USB-audio interface (Focusrite Scarlett 2i2) and were stored as wav-files with a sampling frequency of 22.050 kHz. For the audio–visual text presentation and the recording of the participants’ verbal responses, we used the *Universal Data Acquisition Program* (UDAP; [60]).

Table 2 lists 14 specific diagnostic variables included in this study (7 speech, 7 nonspeech) and short descriptions of how they were assessed and evaluated. Table A1 in Appendix A provides an overview of the BoDyS parameters. The acronyms in the leftmost columns of the tables will be used as shortcuts to denote the assessment parameters throughout the article.

#### 2.2.1. BoDyS Scales

The *Bogenhausen Dysarthria Scales* (BoDyS; [7,61]) were used as a reference standard describing the auditory–perceptual profiles of the participants. The BoDyS comprise nine 5-point scales (0 = profound impairment, 4 = unimpaired) to quantify deficits of respiration, voice, articulation, resonance, and prosody (see Appendix A, Table A1). The BoDyS assessment is exclusively based on speech tasks.

A short version of the BoDyS was used to save testing time. Participants completed a connected speech task comprising six sections of a reading text (13 intonation phrases (74 words, 141 syllables; see Appendix B, Table A3). Stimulus presentation and audio recordings were made according to the audio-supported reading protocol described above. Recordings were evaluated according to the standard BoDyS protocol [7,61].

#### 2.2.2. Articulation Rate (RATE)

Articulation rates were measured from the speech samples of the audio-supported reading task used in the BoDyS assessment (Appendix B, Table A3). The duration of each sentence was measured from the speech oscillogram and spectrogram using Praat [62], and the durations of pauses exceeding 200 [ms] were subtracted from the total utterance duration [63]. In cases where participants omitted or added syllables or words, the syllable count of the text was adjusted correspondingly. The articulation rate (parameter RATE) was calculated by dividing the number of spoken syllables by the duration of the speech sample minus pause durations [syllables per second]. Note that the BoDyS provides ratings of perceived articulation tempo, termed TEM (Appendix A, Table A1), which should not be confused with the acoustic parameter RATE.

#### 2.2.3. Overall Duration of Inspiration Pauses (INSP)

To obtain a measure of speech breathing impairment, audible inspiration pauses in the speech samples of the reading text of Appendix B, Table A3 were labeled in the Praat speech editor [62]. Only inspiration pauses within or between the intonation phrases of each of the six passages of the reading text were considered. INSP indicates the total duration of breathing-related speech pauses (in [s]) added across the six sections of the text.

#### 2.2.4. Overall Duration of Non-Respiratory Filled or Unfilled Pauses (PAUS)

To obtain a measure of speech fluency independent of speech breathing, pauses longer than 200 ms [63] were labeled as filled or unfilled non-respiratory pauses. The sum of all pauses (in [s]) across the speech samples of the reading text, termed PAUS, served as a respiration-independent measure of speech fluency.

#### 2.2.5. Accuracy of Segmental Articulation in Isolated Word Production (WORD)

As the BoDyS articulation scale provides only rather general information about articulatory impairment, a single word repetition task based on 24 near-minimal word pairs was designed to provide more specific evidence (Appendix B, Table A4). The pairs contrasted five consonant features in word-initial and medial position (oral-nasal, voiced-voiceless, alveolar-labial, alveolar-velar, plosive-fricative) and one vowel feature (rounded–unrounded), with four pairs per contrast. Stimulus presentation and audio recordings were made following the audio-supported reading protocol described above. Accuracy of articulation was evaluated for the respective target phonemes using a 3-point scale (0: severely distorted, omitted, or substituted; 1: mildly distorted or prolonged; 2: correct). The parameter WORD specifies the proportion of the maximally attainable score (48 [words] × 2 [maximal points] = 96).

#### 2.2.6. Accuracy of Segmental Articulation in Isolated Syllable Production (SYLL)

A syllable repetition task based on 10 CV-syllables with the vowel /a/ and a selection of consonants (Appendix B, Table A5) was used to assess articulation in greater phonetic detail. Each consonant was rated for the accurate realization of place, manner, voicing, and nasality using a 3-point scale (0: feature not realized; 1: feature ambiguously realized; 2: feature clearly and unambiguously realized). The parameter SYLL specifies the proportion of the maximally attainable score (10 [items] × 4 [features] × 2 [maximal points] = 80).

#### 2.2.7. Intelligibility (INT)

Intelligibility measurements were based on a sentence transcriptions task. Each participant produced one of nine blocks of 10 natural sentences each from an adapted version of the “Marburger Verständlichkeitstest” (Marburg Intelligibility Test) [64,65]. The sentences of this test are between 5 and 12 syllables long, and blocks are balanced for sentence length, syntactic structure, and phonological complexity. The allocation of sentence blocks to participants was pseudo-randomized. Stimulus presentation and the recording of the participants’ responses followed the audio-supported reading protocol described above. The recordings were cleaned of noise, examiners’ utterances, and participants’ comments and adjusted for loudness differences using Praat [62]. Furthermore, a master transcript was prepared for each spoken sentence, in order to adjust for erroneous changes in the wordings of the target sentences.

Thirty listeners (native German speakers, aged 18–58 years with a mean age of 35 years, and 15 women) were involved in the sentence transcription tasks. All listeners were lay persons, that is, they had no professional or private experience with persons with speech impairments. For reasons of project scheduling, listeners were subdivided into three cohorts of 10 listeners each. To avoid familiarity effects, each listener from a cohort transcribed 90 different sentences produced by 90 different speakers, so that each speaker’s speech samples were transcribed by 10 listeners from the same cohort. Each listener was allocated a compilation of speech samples from neurotypical speakers and speakers from all etiologic subgroups in a pseudo-randomized order. Listener sessions started with a fixed set of four practice trials.

Listener sessions were completed individually in a silent laboratory room using a Logitech USB-headset H390. Each sentence could be listened to only once and had to be transcribed orthographically on a protocol sheet. Listeners were instructed to transcribe each word or word fragment they believed to have understood or guessed, or otherwise replace unintelligible parts by dashes. Listener sessions took between 30 and 45 min and were interrupted by a short break after 45 stimuli.

Listener transcripts were compared with the corresponding master transcripts. The parameter INT indicates the proportion of correctly transcribed syllables per participant across the 10 test sentences.

#### 2.2.8. Naturalness (NAT)

Speech naturalness was assessed using the recordings made for the BoDyS analyses (Appendix B, Table A3) as materials. From each participant’s recordings, the first two sections of the text were extracted (25 words, 4 intonation phrases) and adjusted for loudness differences using Praat [62]. For project scheduling reasons, participants were grouped by etiology to form five cohorts of participants, with the PD and PSP patients allocated to one cohort. Each cohort was supplemented by control participants and by three patients each from the other etiologic groups, resulting in cohorts of between 48 and 67 speakers. The speech samples of the cohorts were evaluated by groups of 16 listeners each (native German speakers, 18–60 years old, mean: 33 years; 42 women). All listeners were unfamiliar with impaired speech, and none of the listeners were also involved in the intelligibility assessment.

Listeners were presented with the speech samples of their assigned cohorts in randomized order within individual sessions. Before the experimental items were presented, listeners heard seven warm-up samples of the same materials spanning a large range of dysarthria types and severities, in order to minimize set effects. Listeners rated each sample on a five-point scale according to how natural the utterance sounded to them (5: natural; 1: very unnatural; cf. [9]). The naturalness parameter NAT was determined by averaging the 16 naturalness ratings per participant.

#### 2.2.9. Diadochokinetic Syllable Rate (BABA, DADA, BADA)

The experimental procedure comprised two non-alternating sequences (/baba…/, /dada…/), and one alternating sequence (bada…/). Participants were instructed to repeat each syllable sequence as fast and as accurately as possible on a single breath. Each subject had a demonstration by the examiner and one warm-up trial on each task before two experimental runs were recorded per sequence. For each run, we analyzed a maximum of 10 and a minimum of six syllables, uninterrupted by pauses. Trials that contained fewer repetitions, substitutions, or sequential errors were discarded from further analyses. The duration of each trial was measured using Praat [62]. The mean rate (in syllables per second) was calculated for each trial, resulting in six-syllable repetition rate variables. The better of the two trials of the same task was then used as DDK_syl_ variable, referred to as BABA, DADA, and BADA, respectively. 

The syllable repetition protocol used here deviates from the commonly used standard which is based on repetitions of /pa/, /ta/, /ka/, and /pataka/ [39]. In our experience, the use of aspirated plosives, that is, /p/, /t/, and /k/, in syllable DDK increases the speed–accuracy conflict that is intrinsic to this task type: Since high repetition rates are typically accompanied by hypo-articulated consonant production, that is, a reduction of aspirated to unaspirated plosives or even to flaps or glides, participants in a DDK_syl_ task must choose between preserving plosive aspiration at the expense of speed or maximizing speed at the expense of accuracy. This ambiguity causes high inter-subject variability. As German /ba/ and /da/ involve unaspirated plosives, their use mitigates this conflict and thereby reduces variability. Three-syllabic /badaga/ (or /pataka/) was avoided, because this condition regularly creates sequencing errors or task abortions, especially in participants who prioritize the maximum rate criterion over accuracy. Such breakdowns may reflect an executive control problem more than a motor speech problem [38]. The syllable /da/ was preferred to /ga/ because it allowed us to compare fast repetitions of /da/ with oral–facial nonspeech movements involving the tongue tip.

#### 2.2.10. Diadochokinetic Lip- and Tongue Movement Rate (LPLP, TGTG)

These tasks differed from the syllable DDK tasks in that only single articulator movements without active involvement of laryngeal or respiratory activity were instructed. Single articulator DDK tasks are, for example, included in oral apraxia tests such as the nonverbal oral agility test of the Boston Diagnostic Aphasia Examination (BDAE) [66], which scores the number of repetitions of tongue or lip movements within 5 s, but also in standard dysarthria assessments (e.g., [3]).

In the present study, participants were asked to (a) repeatedly purse and spread their lips and (b) repeatedly move their tongue from the left to the right side of the mouth corner. They were instructed to perform these movements as fast as possible, but still try to reach the target points, and were interrupted after 7–10 movement cycles. Each subject had a demonstration by the examiner and a warm-up trial on each task before two runs were recorded in reverse order. All responses were recorded with a digital video camcorder and analyzed using the annotation program ELAN [67] which allowed the precise localization of the start- and endpoints of a movement cycle. A maximum of four and a minimum of three cycles per task were analyzed using the slow-motion function of ELAN. Trials containing less than three cycles, movement substitutions, or sequential errors were discarded from the analysis. For each trial, the mean rate was calculated in cycles per second, and the better of the two trials per task was selected as variable LPLP and TGTG, respectively. The shortcut DDK_LT_ will be used to distinguish the single articulator DDK tasks involving lip and tongue movements from the syllable repetition (i.e., DDK_syl_) tasks.

#### 2.2.11. Accuracy of Single Nonspeech (Non-Repetitive) Oral-Laryngeal Motor Actions (SNGL)

This type of task is regularly used in standard neurologic examinations of cranial nerve functions, mostly with a focus on detecting postural or movement asymmetries [68], and in oral apraxia tests, usually with a focus on movement errors or inaccuracies [69]. However, these tasks also play an important role in standard dysarthria tests, with their results usually being conflated with those of speech and other nonspeech tasks (e.g., [3,8]).

Twenty-four motor actions involving movements of the lips, jaw, tongue, and the larynx, with or without respiratory activity, were administered (Appendix B, Table A6). Each action was instructed verbally and presented visually by the examiner. The presentation of the models was standardized during examiner training. Trials in which the participant responded synchronously with the examiner were repeated. All responses were recorded with a digital video camcorder and analyzed using the annotation program ELAN [67]. In case of repetitive trials and/or self-corrections, only the first complete trial was analyzed. Evaluations were performed using the slow-motion function of ELAN. In sound-generating tasks, the audio signal was also taken into consideration. Each realization was scored on a 3-point scale (0: severely impaired, e.g., only minimal movement amplitude or parapraxic movement; 1: asymmetric realization, reduced movement amplitude, weak sound generation; 2: unimpaired). The parameter SNGL represents the proportion of maximally attainable scores (24 [items] × 2 [maximal points] = 48).

#### 2.2.12. Maximum Phonation Time (MPT)

This parameter type has been used in different variants, mostly in the form of a sustained vowel task targeting the interplay of phonatory and respiratory functions under maximal demands [48]. In this study, participants were instructed to produce the vowel /a/ as long as possible after taking a deep inhalation. The task was demonstrated by the examiner and a warm-up trial was provided, before two experimental trials were recorded. The duration of each trial was measured manually using the speech editor of Praat [62], and the better of the two trials was used as a measure of maximum phonation time, termed MPT, in seconds.

#### 2.2.13. Reliability

For the auditory-perceptual evaluation of the BoDyS scales, reliability measures were reported earlier [7,61]. According to published data covering more than 7400 rating pairs, interrater agreements per item and scale were in 93% to 98% within one scale point.

A re-evaluation of interrater agreements for the BoDyS scales was made in the present study for a selection of 48 patients and 12 control participants (23% of the total sample) by having each score, as determined by one of the five examiners, be re-evaluated by one of the four other examiners. The same re-evaluation method was used for all parameters except intelligibility (INT) and naturalness (NAT; see below). Krippendorff’s alpha [70] with 95% confidence intervals based on 2000 bootstrapping cycles were obtained using the R-function kripp.boot [71]. The results for the BoDyS scales were good to excellent, with the exception of the pitch-loudness-scale (VPL) which had only moderate reliability (cf. Appendix A, Table A2). For the remaining speech and nonspeech diagnostic parameters, reliability analyses revealed almost exclusively excellent alpha-coefficients, with the exception of PAUS and SYLL (Table 3).

The reliability of the intelligibility scores was ensured by comparing the three listener cohorts for their evaluations of 16 overlapping participants, that is, 160 speech samples evaluated by the three listener cohorts. The scores per participant, averaged across the 10 listeners of each cohort, were correlated significantly between pairs of cohorts (Spearman-ρ ≥ 0.88). A Friedman test revealed no significant differences between cohorts (χ^2^ (2) = 3.00, *p* = 0.22). Averaged across the 16 speakers, the largest intelligibility difference between the two cohorts was 0.02. Krippendorff’s alpha revealed excellent agreement between listener cohorts (Table 3).

The reliability of the naturalness ratings was analyzed across all five cohorts of speakers and listeners. Krippendorff’s alpha revealed a high between listener agreement (Table 3).

### 2.3. Statistics

RStudio 2022.07.1 was used in all statistical computations [72].

#### 2.3.1. Regression-Standardization for Age and Sex

Each parameter was standardized for its variation in the control group and for possible age- and sex-influences. To this end, the lm-function of the R-package ‘stats’ [73] was used to fit linear regression models to the CTRL data for each variable, with age as a continuous variable (see Appendix C, Figure A1) and sex as regressors. Regression residuals characterized each participant’s individual distance from their age- and sex-related reference in the CTRL group. For parameters that were normally distributed in the CTRL group, standard (0, 1) z-scores were computed relative to the distribution of CTRL values, that is, distances from the CTRL mean (=0) in units of one standard deviation. In non-parametric cases, individual scores were standardized relative to the median value of the CTRL distribution, and the difference between the median and the 1st percentile was used to determine a dispersion measure. The wide margins of the distributions of the control data, i.e., 1% instead of the more common 5% threshold were used to determine dispersion units in the non-parametric cases, because some of the variables (e.g., intelligibility) had particularly high kurtoses. In some of these cases, the distance of the 5th percentile from the median was close to 0, and would therefore have inflated the standard scores excessively, whereas an extension to the 1st percentile yielded more manageable distance values.). More specifically, the dispersion unit *d*_0_*^lo^* of the lower tail of the distribution was determined as
*d*_0_*^lo^* = (*q*_0.01_ − *m*)/2.33(1)
where *q*_0.01_ denotes the first percentile and *m* is the median of the distribution of the respective parameter residuals in the CTRL group. The factor 2.33 was chosen to make the non-parametric distance units roughly comparable to a parametric standard deviation unit insofar as the 1% cutoff of a normal distribution has a distance of 2.33 standard deviations from the mean. Thus, comparable standard metrics were provided for all diagnostic parameters.

In all parameters except articulation rate, only the lower tail of the CTRL distribution was relevant. For the parameter RATE, both decreased and increased scores could occur as indicators of impaired performance, so a separate dispersion unit for the upper tail of the distribution had to be determined analogous to Equation (1), but with the 99% cutoff of the CTRL data as the reference score. Parameters expressing proportions (e.g., INT, the proportion of correctly transcribed syllables) were cosine-transformed before standardization.

#### 2.3.2. Exploratory Graph Analyses (EGA) and Confirmatory Factor Analyses (CFA)

In order to analyze the dimensionality of the parameter space and the structural relationships between variables, Exploratory Graph Analyses (EGA) were performed for standardized variables over the patient data set (*n* = 130). EGA is based on methods developed in network psychometrics to study psychological constructs by estimating the number of latent variables underlying multivariate diagnostic data [51]. It is more robust than other factor analysis methods and particularly suitable for sample sizes lower than 500. To validate the exploratory models, confirmatory factor analyses (CFA) were calculated using the EGA dimensions as factors and the associated diagnostic parameters as indicators. Model fitting was based on Maximum Likelihood Estimation and used the R-package *lavaan* [74]. Goodness-of-fit was assessed using the *Root Mean Square Error of Approximation* (RMSEA) as a parsimony-corrected index measuring the extent to which the model reproduces the variances and covariances of the sample, and the *Comparative Fit Index* (CFI) and the *Tucker-Lewis Index* (TLI) as measures of how much better the model fits the data compared to a “null model”. According to conventions, RMSEA indices ≤0.08 or ≤0.05 suggest “adequate” or “good” model fit, respectively, and models with RMSEA ≥0.10 should be rejected. CFI and TLI values between 0.90 and 0.95 indicate “acceptable”, and values >0.95 “good” model fit [75].

## 3. Results

### 3.1. Parameter Distributions

Table 4 presents an overview of the distributions of BoDyS raw and standard scores in the control participants and the patients. Recall that all variables were normalized for age and sex. Significant influences of age were present in the respiration (RSP) and the voice stability raw scores (VOS), a significant influence of sex was present in voice quality (VOQ).

There was overlap between the PAT and the CTRL participants on all scales, but completely unremarkable profiles were rare in the patients. The most severe impairments were observed in articulation (ART), prosodic modulation (MOD), articulation rate (TEM), and respiration (RSP). Student’s *t*-tests revealed significant differences between the two groups in all parameters.

Figure A2 in Appendix A plots the distributions of standardized BoDyS total scores in both groups (see also Table 4, bottom row), illustrating that the distribution in the patient group was strongly left-skewed, that is, the vast majority of the neurologic patients (94%) ranged clearly below the median of the CTRL group. Overall, 91/130 (70%) had BoDyS total scores below the 5th percentile of the controls, and 79/130 patients (61%) were below the 1% cutoff, suggesting that between 60 and 70% of the patients most likely had dysarthria.

Figure 1 depicts the distributions of the specific speech- (top panel) and nonspeech parameters (bottom panel) in the patient sample. Standard scores are plotted for all parameters, with the dashed horizontal lines delineating the 1–99% range of the control data. Table A7 in Appendix C documents the raw and ƶ-scores corresponding to Figure 1. Student’s *t*-tests revealed significant group differences in all parameters.

Notably, in the patients, most of the standardized speech parameters covered a much wider range than the nonspeech parameters, which was due to the narrower distributions of speech parameters in the neurotypical participants (Figure 1). The clearest separation between the CTRL and PAT groups resulted in the naturalness ratings, whereas for other variables, considerable overlap was found, for example, in the perceived accuracy of segmental articulation in isolated words and syllables and in speech pauses, as well as in repetitive lip (LPLP) and tongue movement rates (TGTG) and in maximum phonation times (MPT).

### 3.2. Factor Structure and Dimensionality of the Assessment Parameters

To explore how the speech and nonspeech parameters interacted and thereby to cross-validate nonspeech by speech measures, we applied Exploratory Graph Analysis (EGA; R-function boot EGA [76]) to fit the standardized scores of all diagnostic variables across the patient group. A graphical LASSO model was computed using the walktrap algorithm with 1000 bootstrap iterations. The “Leading Eigenvalue” (LE) algorithm was applied to the Spearman correlation matrix. For details regarding this method see [53].

The EGA model converged to a 5-factor solution in 100% of the bootstrap iterations, demonstrating a high dimension stability. Figure 2 (top) plots the graphical solution and illustrates how the diagnostic variables were allocated to the five dimensions. The nodes represent the diagnostic variables; edges represent the strengths of the connections between variables (essentially partial correlations). The bottom panel of Figure 2 displays the EGA loadings of each dimension on each variable, with high loadings indicating a strong association with the respective factor.

Dimension 1 (red nodes/bars in Figure 2) comprised the BoDyS resonance, articulation, articulation rate, and prosodic modulation scales (RES, ART, TEM, MOD), as well as segmental accuracy in word- and syllable production (WORD, SYLL), together with intelligibility and naturalness (INT, NAT). Particularly strong bindings connected naturalness with ART, TEM, and MOD as well as the BoDyS articulation scale (ART) with the two syllable- and word-based articulation parameters (WORD, SYLL) and with intelligibility (INT). As all indicators of dimension 1 were speech parameters, this dimension will be referred to as SPEECH 1.

Dimension 2 (orange nodes/bars in Figure 2) comprised parameters related with speech fluency, that is, the BoDyS fluency scale (FLU) and the acoustic measures of speech pauses and articulation rate (PAUS, INSP, RATE), as well as BoDyS respiration parameter (RSP). Low RSP scores are usually associated with increased speech breathing, which in turn has an impact on fluency [77]. Particularly strong connections were held between perceived fluency (FLU) and speech pauses (PAUS) and between inspiration pauses (INSP) and articulation rate (RATE). As all indicators of dimension 2 were speech parameters, it will henceforth be referred to as SPEECH 2.

Dimension 3 (yellow nodes/bars in Figure 2) comprised the three voice scales of the BoDyS and will be referred to as SPEECH 3.

Dimension 4 (blue nodes/bars in Figure 2) comprised the three DDK_syl_ rate variables BABA, DADA, and BADA, with a strong binding between BABA and DADA. Note that this dimension had very low loadings on all speech parameters except RATE.

Dimension 5 (grey nodes/bars in Figure 2) comprised the three nonspeech oral motor tasks, that is, the alternating lip and tongue movement rates (LPLP, TGTG) and the single nonspeech oral motor accuracy measure (SNGL), together with maximum phonation time (MPT). This dimension will be referred to as NSOM/MPT. Its loadings on the speech parameters were low. Note that MPT had low or zero loadings from all five dimensions, indicating an outsider role of this measure within the set of diagnostic parameters examined here.

A notably strong binding between dimensions was found between perceived articulation rate (TEM; SPEECH 1) and articulation rate measured in the acoustic signal (RATE, SPEECH 2). Note that the three SPEECH dimensions had very low or even zero loadings on the DDKsyl and the NSOM/MPT parameters.

To validate the EGA model, a confirmatory factor analysis (CFA) with Maximum Likelihood estimation was performed, including the five EGA dimensions as factors and their corresponding parameters as indicators. The metric of the five latent variables was determined by fixing their variances to 1.0. The model was overidentified with 220 degrees of freedom. Admittedly, with 23 indicators and a sample size of 130, this analysis can only be seen as orienting, as it bears the risk of overfitting. The CFA model fitted the data with acceptable overall fit statistics: The root mean square error of approximation (RMSEA) was 0.08, which suggests adequate fit, and comparative fit indices of CFI = 0.90 and TLI = 0.88 were indicative of acceptable model fit (cf. [75]. The correlations between the five dimensions ranged between r = 0.37 (SPEECH3 vs. NSOM/MPT) and r = 0.81 (SPEECH2 vs. DDK). The complete correlation matrix is plotted in Table A8 of Appendix D. Inspection of the CFA loadings revealed an area of strain in the NSOM factor with a loading of only 0.35 on maximum phonation time (MPT), which underscores the outsider role of this parameter and explains why the model fit was only moderate.

### 3.3. Effectors, Domain–General Functions or Tasks?

As mentioned in the introduction, one of the reasons why nonspeech tasks are so common in dysarthria assessment is that they can be designed for high effector-specificity [15]. Another hypothesis was that nonspeech tasks can be designed to test specific functional goals on which they overlap with speech [16]. The EGA model of Figure 2 is inconsistent with the effector-specificity assumption insofar as the lip- and tongue-related DDK-parameters were not assigned to separate lip and tongue dimensions, respectively, but were grouped by task, that is, syllable DDK (BABA, DADA) and single-articulator DDK (LPLP, TGTG). Furthermore, the result that two parameters reflecting fundamentally different functional goals, that is, maximum repetition rate (LPLP, TGTG) and single movement accuracy (SNGL) were allocated to the same factor NSOM/MPT, whereas the maximum repetition parameters with their overlapping functional goals were split into two separate factors, that is, NSOM/MPT and DDK_syl_, is inconsistent with the function-specificity assumption.

Because the analysis presented in the previous section did not consider this aspect systematically enough, potential evidence for effector- or function-specificity may have gone unnoticed. In particular, the EGA model was not specified for effector-specific speech- and single-articulator tasks. To refine the model for suitable testing of the three competing assumptions, that is, effector-, function- or task-specificity, the speech parameters describing the accuracy of segmental articulation in words and syllables (WORD, SYLL) was re-analyzed by splitting the materials into items involving lip vs. anterior tongue movements, that is, labial vs. coronal consonants (see Appendix B, Table A4 and Table A5). Furthermore, the nonspeech parameter characterizing the accuracy of single articulator movements (parameter SNGL) was split into items addressing the motility of the lips and the tongue separately (Appendix B, Table A6). Together with the two-syllable repetition parameters BABA vs. DADA and the two repetitive lip and tongue movement tasks LPLP vs. TGTG, this design comprised four effector-specific speech parameters, that is, WORD.LP/SYLL.LP and WORD.TG/SYLL.TG for the accuracy of labial and alveolar consonant articulation, respectively, and six effector-specific nonspeech parameters. The latter were split into two parameters characterizing the accuracy of non-repetitive single articulator movements, referred to as LP and TG, two parameters characterizing the maximum speed of repetitive single articulator movements (LPLP and TGTG), and two maximum repetition rate parameters for the lips and the tongue blade (BABA, DADA). This selection of diagnostic parameters allowed us to analyze, along with effector-specificity (labial vs. lingual), also the function-specificity of speech and nonspeech parameters, that is, whether they test the involved functional goals, accuracy vs. speed, across the speech and the nonspeech domains. The parameter set was further expanded by the effector-unspecific intelligibility parameter INT. In an effector-specific diagnostic model, we expected that INT would be excluded from the lip and tongue dimensions. In a function-specific model, it would be assigned to the accuracy- rather than the speed dimension and in a task-specific model, it would go together with the WORD- and SYLL-parameters.

Figure 3 depicts an EGA-model of these parameters over the patient sample (*n* = 130), with the graphical solution (top) and the EGA loadings (bottom). The model converged to a 3-dimensional solution with no variation of dimensionality across 1000 bootstrap iterations, indicating high dimensional stability of the solution.

A first dimension, termed SPEECH, encompassed the four parameters representing accuracy of consonant articulation together with intelligibility, that is, exclusively speech-related parameters. A second dimension aggregated the repetitive and non-repetitive single articulator parameters (NSOM), and a third dimension combined the two-syllable DDK parameters. Thus, the three factors respected neither effector- nor function boundaries: the lip and tongue-related parameters were all conflated within dimensions, the parameters targeting speed goals were split between the NSOM and the DDK_syl_ dimensions, and the parameters targeting spatial accuracy were split between the SPEECH and the NSOM dimensions. In all dimensions, strong bindings were found between the lip- and tongue-specific variables within parameter types, suggesting that it was the characteristics of the task rather than the particular effector which made parameters structurally similar.

To validate the exploratory model, a confirmatory factor analysis (CFA) was performed with the three EGA-dimensions as factors and their corresponding parameters as indicators. Given the inclusion of only 11 indicators with a sample size of 130, the risk of overfitting was lower with this smaller measurement model [75]. The model fitted the data exceptionally well (RMSEA = 0.02, CFI = 0.99, TLI = 0.99; cf. Section 2.3). For the intercorrelation matrix of the three factors see Table A9 of Appendix D.

To further test the assumptions of effector-, function-, and task-specificity against each other, a simplified, orthogonal design based on one pair of effector-specific variables per parameter type was included, and intelligibility was dropped from the parameter list (see Table 5). As for the effector hypothesis, a confirmatory factor analysis was designed using a two-factor measurement model in which the four lip-related variables were modeled to load onto a LIP factor, and the four tongue-related variables onto a TONGUE factor. The two latent variables were permitted to be correlated, based on the assumption that external variables may exert third-variable influences on both factors. The fit indices of this model indicated an extremely poor fit, with CFI and TLI coefficients substantially below 0.90 and an RMSEA coefficient far above 0.10 (Table 5). Therefore, the effector-model was abandoned (for goodness-of-fit criteria cf. Section 2.3).

A further two-factor measurement model with domain-general functions as latent variables was designed, in which the two speech and the two single, non-repetitive articulator variables were modeled to load onto an ACCURACY factor and the four maximum repetition rate parameters onto a SPEED factor. Again, the two factors were permitted to be correlated. The fit indices of this model also indicated a poor fit, with CFI and TLI coefficients clearly below 0.90 and an RMSEA coefficient clearly above 0.10 (Table 5), leading to the rejection of the model based on domain–general functions.

In a third CFA model, the task hypothesis was tested again for the more selective parameter ensemble of Table 5. The model was designed for three latent variables, that is, SPEECH, NSOM, and DDK_syl_, with diagnostic parameters assigned as indicators as shown in Table 5. The comparative fit indices of this model, CFI and TLI, were both clearly above 0.95, and the RMSEA was 0.08, indicating a fully adequate model fit.

As a result, the hypothesis of task-specificity was maintained, whereas the two alternative models of effector- and function-specificity were abandoned.

## 4. Summary and Discussion

This study was planned to address a problem raised by Folkins and coworkers in a highly recognized article, that is, whether there is a role for nonspeech parameters in the assessment of the speech impairment of patients with motor speech disorders, especially dysarthria [15]. To our knowledge, this is the first systematic and comprehensive study that attempts to answer this question by systematically testing the validity of commonly used nonspeech parameters relative to auditory-perceptual and acoustic speech parameters in standard clinical assessments of dysarthria.

A basic assumption of this approach was that the use of nonspeech parameters in speech assessment calls for extra validation efforts because they lack any face validity as direct measures of speech impairment. Maximum performance measures or parameters describing the accuracy of oral–facial movement imitation do not reflect the complaints patients experience in their daily verbal interactions, and their improvement is not among the primary goals of therapeutic interventions. Therefore, the validity of nonspeech measures as diagnostic markers of dysarthric impairment needs to be established empirically. In contrast, each of the speech parameters reported here tells something that is potentially relevant with regard to a patient’s speech impairment, for example, their articulation rate and speech pauses; their precision in consonant articulation; or their profile of respiratory, vocal, articulatory, and prosodic deficits, and each of them may directly contribute to intelligibility problems, unnatural speech, or increased speech effort. Thus, there is a clear imbalance between speech and nonspeech parameters as far as their face validity as diagnostic measures of dysarthria is concerned. In addition, unresolved theoretical issues compel us to challenge the diagnostic value of nonspeech parameters in the clinical assessment of a patient’s motor speech impairment. The most general theoretical question is to what extent the organization of motor behavior differs between speaking and other oral–facial motor activities [23,35,78], which in turn is related to questions about the role of practice-dependent brain plasticity, as well as the particularities of motor goal setting and of sensorimotor principles in specifying speech motor functions over other motor functions of the speech musculature [17,18,19,24,29].

### 4.1. Parameter Distributions

Overall, the patients’ standardized nonspeech scores apparently deviated less from the control data than their speech scores. Despite the fact that all participants were given the same standardized instructions, there was a wide variation in most nonspeech parameters among control participants, most remarkably in maximum phonation time and in the maximum repetition rates of single articulator movements. The scores achieved by even the more severely impaired neurologic patients were not much lower than the cutoff scores of the CTRL group in these parameters. In contrast, some of the speech parameters, especially intelligibility, were very narrowly distributed in the controls relative to the patients, leading to large deviations of the standard scores in patients with severe speech impairment. These results underscore the importance of using standardized scores if comparisons between diagnostic parameters are made.

### 4.2. Factor Structure of Speech and Nonspeech Diagnostic Parameters

The exploratory graph analysis described in Section 3.2 provided a complete picture of the structural relationships between the speech and nonspeech diagnostic parameters included in this study. The most important outcome was that the EGA model separated sharply between the speech and the nonspeech diagnostic parameters. The decomposition of the speech-related diagnostic variables into three latent traits followed a plausible pattern: In the first dimension (SPEECH 1), the two communication-related parameters INT and NAT clustered with parameters describing articulation, resonance, and prosody, SPEECH 2 covered predominantly rate and fluency parameters, and SPEECH 3 covered the three BoDyS voice scales.

The disconnection of the three DDK_syl_ parameters from the speech parameters in the EGA model confirmed earlier results obtained in the same patient sample using a different methodological approach and a smaller set of diagnostic variables [45]. In particular, the three syllable repetition parameters had nothing to do with communication-related measures, that is, intelligibility and naturalness, with accuracy of articulation, voice parameters, or prosody. They were also separated from the speech rate and fluency dimension (SPEECH 2), although the CFA factors corresponding with these two dimensions were highly correlated. These findings reinforce the need to distinguish between how quickly persons with dysarthria can repeat a given syllable and how they produce words and sentences under largely unconstrained conditions. The diagnostic relevance of this distinction obviously derives from the different motor organizing principles that underlie these two types of actions, as discussed in the introduction (Section 1.2.1) [17,18,19]. Moreover, DDK_syl_ rate relies on cognitive functions [38] and, as a maximum performance task, on vigilance and the motivation to achieve the best possible performance, which can be particularly relevant in neurologic populations [32]. Taken together, these considerations suggest that syllable repetition at maximal speed challenges not only the adaptability of the motor system to cope with novel task demands, but also the executive control functions involved in cognitive switching and, not to forget, the motivation and drive to do one’s best at a strenuous task during the course of a sometimes tiring diagnostic session.

Maximum syllable repetition rates were disconnected not only from all speech parameters, but also from the three NSOM parameters describing rate and accuracy of single articulator movements. These parameters obviously have commonalities which distinguish them from both speech and syllable DDK. In particular, the accuracy of single articulator movements, as assessed by the parameter SNGL, had nothing to do with consonant articulation in word- or syllable production, with the BoDyS articulation score, or with intelligibility. Articulation in speech and isolated articulator movements in gesture imitation obviously recruit the same effector systems in different ways. Likewise, the rates of repetitive lip and tongue movements were unrelated to repetition rates of the same articulators in the /ba/ and /da/ tasks, that is, the motor goals of repeating a single articulator movement or an entire syllable at maximum rate appeared to recruit the articulatory motor subsystems in different ways. These points will be addressed in greater detail in the following paragraph.

### 4.3. Task-Specificity Dominates Effector- and Function-Specificity

Central to the tradition of using nonspeech tasks in dysarthria diagnostics is the assumption that they can be designed more specifically than speech tasks to isolate the contributions of different vocal tract components [15] or of domain–general motor functions [16] to dysarthria. With respect to the data reported here, the assumption of effector-specificity would imply that nonspeech parameters addressing the labial–mandibular and anterior lingual subsystems, respectively, are specifically predictive of problems in labial and lingual consonant articulation. The hypothesis that different oral–facial motor tasks share general functional goals with each other and with speech, as postulated in the integrative model proposed by Ballard and coworkers [16], would predict that parameters representing the goal to achieve maximum repetition rates (i.e., BABA, DADA, LPLP, TGTG) are separated from parameters representing accuracy in attaining specified targets without any speed requirements (i.e., WORD.LP, WORD.TG, LP, TG).

However, the EGA model plotted in Figure 3 and subsequent CFA analyses did not support any of the two assumptions, as they clearly separated variables along task- rather than effector- or function boundaries. It separated parameters into a SPEECH dimension comprising accuracy of consonant articulation and intelligibility, an NSOM dimension encompassing the isolated and repetitive single articulator parameters, and a DDK_syl_ dimension encompassing the two-syllable repetition tasks and collapsed lip- and tongue-related as well as speed- and accuracy-related variables within these dimensions. Alternative CFA models, designed to distinguish between two effector-specific (i.e., LIP vs. TONGUE) and two function-specific factors (i.e., SPEED vs. ACCURACY), respectively, failed to fit the data with any reasonable accuracy.

Thus, irrespective of the involved effector organs, the repetitive and the isolated single articulator movements seemed to imply comparable motor demands. In particular, the motor requirements of the two repetitive NSOM tasks apparently differed from those of the two DDKsyl tasks, despite the fact that they involved the same functional goal of achieving maximum rates in cyclic lip or tongue movements in both cases. On the other hand, the labial/lingual isolated NSOM tasks and the word production tasks involving the realization of labial/lingual target consonants seemed to recruit the involved effector organs in different ways, although they both targeted a similar functional goal, that is, to realize specific movement targets with sufficient accuracy. Contrary to what would be suggested by an integrative model of oral–facial motor control [16], neither the overlapping motor effectors, that is, lips vs. tongue, nor the overlapping functional determinants, that is, speed vs. spatial accuracy, had an effect on the grouping of diagnostic parameters across the speech–nonspeech boundary. Therefore, as predicted by task dynamic theories, it turned out that “the same vocal tract organs may be governed by different dynamics in [different] functions […] and different organs may be governed by similar dynamics in the same function” [23]; p. 349.

Some of the results reported here might be complemented by further statistical analyses. For example, it should be analyzed if the relationships between speech and nonspeech parameters depend on etiology or the type of dysarthric impairment. Earlier data have indicated that such influences exist (e.g., [33]). This finding, if replicated, cannot be reconciled with either the effector-specificity hypothesis or the assumption of domain–general functional goals, because any true functional or effector overlap would cause the same etiology-dependent motor deficits across all modalities. A further question that remains relates to the finding that the DDK_syl_-dimension of the EGA model plotted in Figure 2 was different from, but highly correlated with, the speech dimension combining rate and fluency parameters (see Table A8 in Appendix D). Separate hypothesis-driven analyses are required to further elucidate the specific relationship between maximum syllable repetition rate on the one hand and articulation rate and fluency on the other hand.

## 5. Conclusions

The results of the present study failed to support the idea that nonspeech parameters reflect relevant speech characteristics in patients with neurological movement disorders. In particular, the diagnostic variables examined here decomposed into disjoint factors without any overlap between the speech and the nonspeech domains. This outcome is compatible with earlier experimental and clinical data in suggesting that speech motor control is adapted to the specific goal of creating intelligible, fluent, and naturally sounding words and sentences. According to the evidence gained in this study, the clinical characteristics of impaired speech in PWD cannot adequately be assessed using standard nonspeech oral–facial motor tasks.

Various methodological factors might be invoked to explain this negative result, for example, that the study lacked the required experimental rigor in designing more compatible assemblies of speech and nonspeech parameters or that nonspeech analyses were based on standard evaluation methods rather than on more advanced techniques, such as visuomotor tracking examinations [16] or machine-learning supported DDK-analyses [39]. Yet, our selection of tasks and parameters aimed to represent current clinical standards rather than future perspectives, in order to critically discuss and improve today’s practice. Moreover, our results are underpinned by theoretical arguments that focus on task-specific principles of speech-motor control, suggesting that even more sophisticated methodological approaches would not necessarily change the outcome fundamentally. Nevertheless, new validation studies will be needed as new methods are introduced into clinical care in the future.

From the perspective of current clinical standards, our study revealed no evidence that nonspeech parameters can be helpful in analyzing the motor speech profiles of patients with neurologic disorders. Therefore, in the interest of diagnostic economy and to reduce the burden of lengthy and exhausting diagnostic procedures, we suggest that these tasks be used only when there is credible evidence about how they contribute to clinical decision making; for example, regarding the course of bulbar involvement in progressive diseases, cranial nerve lesions, or specific differential-diagnostic issues that may arise [13]. Thus, nonspeech tasks should be used selectively for well-established purposes rather than routinely in dysarthria assessment, and their results should be interpreted in light of their specific motor, cognitive, and attentional demands and not conflated with speech characteristics.

In clinical research, proponents of the use of nonspeech parameters in dysarthria assessment could strengthen their position by providing empirical evidence for the concurrent validity of nonspeech parameters with characteristics of dysarthric speech. At the same time, more effort should be made to sharpen the specificity of speech-based parameters for differential diagnostic, prognostic, and therapeutic decisions. In particular, advanced speech processing technologies should be made usable to clinicians, in order to more efficiently extract relevant information from naturalistic speech samples and make such information available in standard clinical assessment of dysarthria.

## Figures and Tables

**Figure 1 brainsci-13-00113-f001:**
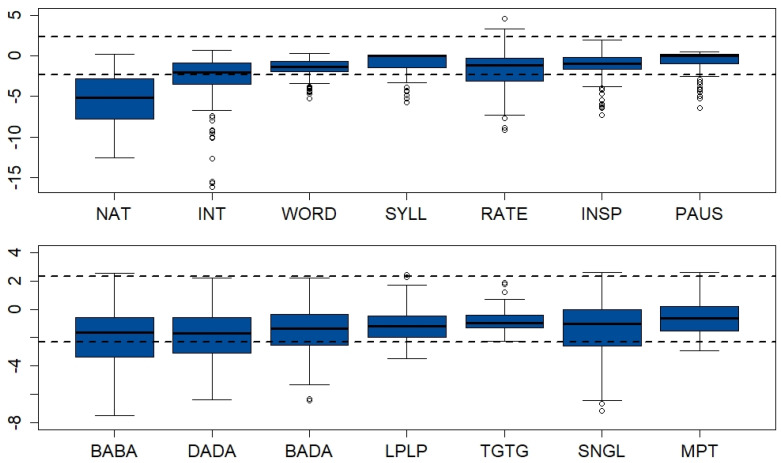
Distributions of standardized speech- (**top panel**) and nonspeech parameters (**bottom panel**) in the patient group (*n* = 130). Dashed horizontal lines demarcate the 1% and the 99% cutoffs of the control participants (*n* = 130). Note the different scale widths of the y-axes in the two panels. NAT: naturalness; INT: intelligibility; WORD: articulatory accuracy of consonants and vowels in word context; SYLL: articulatory accuracy of consonants in syllable context; RATE: articulation rate (acoustic); INSP: total duration of inspiration pauses; PAUS: total-duration of non-respiratory pauses; BABA, DADA, BADA: DDK_syl_ rate (syllables /ba/, /da/, /bada/); LPLP: repetition rate of lip pursing/spreading; TGTG: repetition rate of left–right tongue movements; SNGL: accuracy of oral movement imitation; MPT: maximum phonation time.

**Figure 2 brainsci-13-00113-f002:**
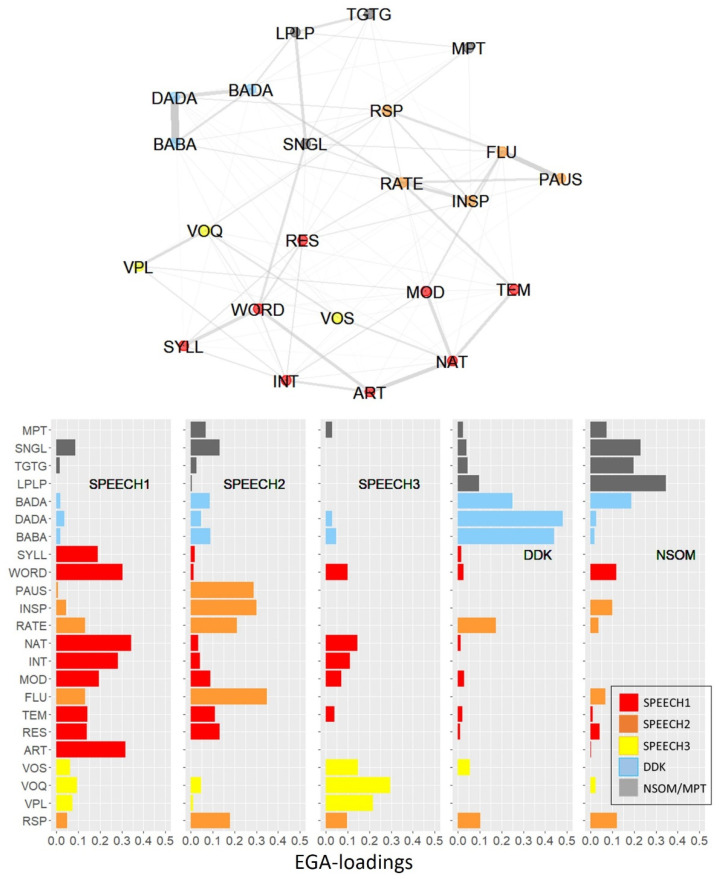
Exploratory graph analysis of all standardized assessment parameters in the patient sample (*n* = 130). (**Top**): graphical solution. (**Bottom**): Loadings of EGA-dimensions on parameters. Red: SPEECH 1; orange: SPEECH 2; yellow: SPEECH3; blue: DDK_syl_; grey: NSOM/MPT.

**Figure 3 brainsci-13-00113-f003:**
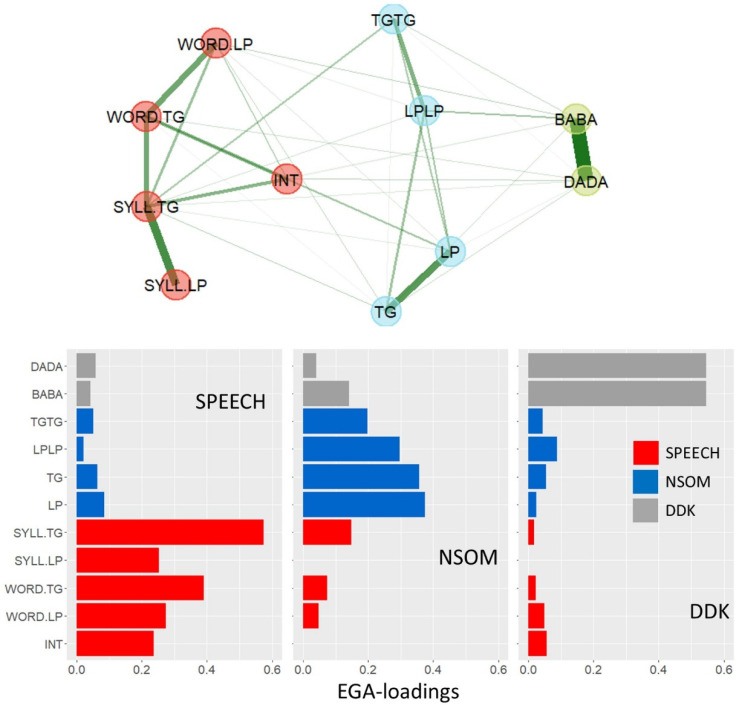
Exploratory graph analysis of assessment parameters specified for the involvement of labial vs. lingual gestures in the patient sample (*n* = 130), together with intelligibility (INT). (**Top panel**): graphical solution; (**bottom panel**): EGA factor loadings. Red: Factor SPEECH; blue: factor NSOM; light green/grey: factor (syllable) DDK. CFA fit indices: CFI = 0.99, TLI = 0.99, RMSEA = 0.02 (“good” model fit; cf. Section 2.3).

**Table 1 brainsci-13-00113-t001:** Etiologic sub-samples of the patient sample.

Subgroup	*n*	Women	Age [Years]	m.s.o. ^1^	BoDyS ^2^
Cerebrovascular accident (CVA)	26	8	57 (37,78)	6 (2,368)	3.2 (1.3,3.9)
Cerebral palsy (CP)	22	6	23 (19,57)	congenital	2.5 (1.6,3.6)
Cerebellar ataxia (ATX)	24	11	55 (35,84)	95 (12,276)	3.3 (1.9,4.0)
Parkinson’s disease (PD)	25	13	72 (46,78)	82 (10,240)	3.6 (2.0,4.0)
Progressive supra-nuclear palsy (PSP)	17	9	69 (60,78)	50 (5,83)	2.9 (1.9,3.9)
Primary dystonia (DYST)	16	7	62 (39,81)	212 (59,450)	3.4 (2.2,3.8)
patients (total)	130	54	60 (19,84)	60 (2,450)	3.3 (1.3,4.0)
CTRL	130	67	49 (18,90)	--	3.9 (3.4,4.0)

^1^ Months since onset. ^2^
*Bogenhausen Dysarthria Scales* (BoDyS) [7], a standardized dysarthria assessment tool based exclusively on speech tasks; raw BoDyS total scores (averaged across 9 scales; 0 = profound impairment; 4 = no impairment).

**Table 2 brainsci-13-00113-t002:** Overview of specific diagnostic parameters (for details see text).

Type		Parameter		Task	Evaluation
		Acronym	Description		
**Speech**		RATE	articulation rate [syll/s]	reading/repetition ^1^	acoustic
		INSP	duration of inspiration pauses [s]	reading/repetition ^1^	acoustic
		PAUS	duration of other pauses [s]	reading/repetition ^1^	acoustic
		WORD	accuracy of segmental articulation	word repetition ^2^	auditory
		SYLL	accuracy of segmental articulation	syllable repetition ^3^	auditory
		INT	intelligibility (for laypersons)	sentence repetition	transcription
		NAT	naturalness (for laypersons)	sentence repetition	rating
**Nonspeech**	DDK_syl_	BABA	syllable repetition rate /ba/ [syll/s]	“on a single breath” ^4^	acoustic
		DADA	syllable repetition rate /da/ [syll/s]	“on a single breath” ^4^	acoustic
		BADA	syllable repetition rate /bada/ [syll/s]	“on a single breath” ^4^	acoustic
	DDK_LT_	LPLP	alternating lip movement rate [cycles/s]	lips: spread-purse ^4^	visual
		TGTG	alternating tongue movement rate [cycles/s]	tongue: left-right ^4^	visual
		SNGL	single oral-facial movement accuracy	action imitation ^5^	visual/auditory
		MPT	maximum phonation time [s]	sustained vowel /a/	acoustic

^1^ Audio-supported text reading (Appendix B, Table A3). ^2^ 24 near-minimal word pairs (Appendix B, Table A4). ^3^ 10 CV syllables (Appendix B, Table A3). ^4^ “maximally fast”. ^5^ 24 single oral–facial or respiratory-laryngeal actions (Appendix B, Table A6).

**Table 3 brainsci-13-00113-t003:** Examiner agreements for all diagnostic parameters. Krippendorff’s alpha across 60 participants (48 patients, 12 controls) and two raters per sample. Reliability analyses for INT and NAT were based on data from listener cohorts consisting of laypersons (see text).

Parameter	Alpha	95% CI	Parameter	Alpha	95% CI
RATE	0.98	(0.96,0.99)	BABA	0.96	(0.87,0.99)
INSP	0.97	(0.94,0.99)	DADA	0.94	(0.86,0.99)
PAUS	0.79	(0.58,0.94)	BADA	0.99	(0.99,0.99)
WORD	0.88	(0.81,0.93)	LPLP	0.99	(0.99,0.99)
SYLL	0.64	(0.39,0.83)	TGTG	0.91	(0.73,0.99)
INT ^1^	0.88	(0.82,0.94)	SNGL	0.95	(0.92,0.97)
NAT ^2^	0.74	(0.70,0.77)	MPT	missing	

^1^ Mean intelligibility scores of 16 participants evaluated by all listeners were compared between the three listener cohorts. ^2^ Listener ratings (*n* = 16 per participant) were collapsed across five cohorts.

**Table 4 brainsci-13-00113-t004:** Raw scores and standardized scores of the BoDyS scales (median and range) for the CTRL and the PAT groups. Standardized CTRL scores (median = 0, 1%-cutoff = −2.33) are not listed.

Scale	CTRL, Raw Scores		PAT, Raw Scores		PAT, ƶ-Scores		*t*-Test ^1^
	Med	Range	Med	Range	Med	Range	df = 258
RSP	4	(2,4)	3	(0,4)	−1.8	(−8.6,0.5)	7.2 ***
VPL	4	(3,4)	4	(1,4)	−0.1	(−7.0,0.1)	7.2 ***
VOQ	3	(2,4)	3	(1,4)	−0.5	(−4.1,1.8)	8.9 ***
VOS	4	(2,4)	4	(1,4)	−0.3	(−5.2,0.6)	6.7 ***
ART	4	(3,4)	3	(1,4)	−2.3	(−7.0,0.1)	12.8 ***
RES	4	(3,4)	4	(1,4)	−0.1	(−7.0,0.1)	6.5 ***
TEM	4	(3,4)	4	(1,4)	0.0	(−7.1,0.1)	7.8 ***
FLU	4	(2,4)	4	(1,4)	0.0	(−4.3,0.1)	7.8 ***
MOD	4	(2,4)	3	(1,4)	−2.3	(−7.0,0.2)	11.9 ***
BoDyS total	3.9	(3.4,4.0)	3.3	(1.3,4.0)	−3.0	(−12.9,1.2)	13.5 ***

^1^ PAT vs. CTRL, standardized scores, one-tailed. ***: *p* < 0.001.

**Table 5 brainsci-13-00113-t005:** Design of three measurement models to explain the covariance structure of lip and tongue motor actions in consonant accuracy during word production (SPEECH; light grey), in isolated and repetitive single articulator movements (NSOM; grey), and syllable DDK (dark grey) through confirmatory factor analyses. (a) Effector model (LIP vs. TONGUE), (b) functional goal model (ACCURACY vs. SPEED), (c) task model (SPEECH vs. NSOM vs. DDK_syl_).

		Domain-General Functions df = 19; CFI = 0.88; TLI = 0.82; RMSEA = 0.17			
		Accuracy		Speed	
**Effectors** **df = 19;** **CFI = 0.82;** **TLI = 0.73;** **RMSEA = 0.21**	**Lip**	WORD.LP	SNGL.LP	LPLP	BABA
	**Tongue**	WORD.TG	SNGL.TG	TGTG	DADA
		**SPEECH**	**NSOM**		**DDK_syl_**
		**Task** **df = 20; CFI = 0.97; TLI = 0.96; RMSEA = 0.08**			

Bold face: Dimensions and associated model fit coefficients.

## Data Availability

The conditions of our ethics approval do not permit public archiving of the study data. Readers seeking access to the data should contact the senior author [A.S.] of the study. Access will be granted to named individuals in accordance with ethical procedures governing the reuse of clinical data upon request and ensuring patients’ anonymity, including the completion of a formal data-sharing agreement and approval of the local ethics committee.

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
