# Peer review of "Speech and Nonspeech Parameters in the Clinical Assessment of Dysarthria: A Dimensional Analysis"

_brainsci, 2023, doi:10.3390/brainsci13010113_

Round 1
Reviewer 1 Report
Thank you so much for the opportunity to review this very interesting and exciting paper. The paper addresses an important and currently understudied research topic. I believe that the study is of high scientific quality based on rigorous experimental methods. It is evident that a lot of effort and care went into the preparation of the manuscript. It is easy to follow and contains all necessary information to allow replication studies. Despite its comprehensive nature, the content of the paper is not overwhelming. The authors have done a remarkable job of presenting the data in a way that is easily accessible to the reader. Well done!
I only have a few suggestions to make that may improve the clarity of the paper. I think the study will be of high interest to the readership of the journal.
abstract:
line 32: consider changing the sentence to "perceptual speech characteristics of dysarthria in patients with neurological disorders".
line 35: consider "auditory-perceptual ratings and speech acoustic measures"
line 37: consider replacing "gender" with "sex" throughout the paper because gender refers to the social construct while sex refers to the biological attribute.
line 38: "were used " instead of "was used"
line 39: The sentence "in a first analysis we tested...." is really difficult to understand because it is so long. Perhaps try to split up into several sentences and rephrase.
line 43: the expression "decompose along ..." is difficult to understand. Perhaps rephrase?
line 45: "diagnostic measures of speech characteristics in PWD" - I have had trouble throughout the manuscript with the terms "measure", "parameter" and "variable" - they seem to be used interchangeably but it would help if the use of these words were more consistent. What is a parameter, what is a measure, and what is a variable?
Introduction
line 51: Suggestion to change to "Dysarthria is a complex neurogenic motor speech disorder"
line 54: As a result of these deficits... - this sentence is difficult to comprehend. Perhaps add "may sound less "natural", and may require more effort and attention from conversation partners"
Line 67: As a signature of impaired speech motor functions, these parameters are essential... - the reference to "these parameters" is unclear and the sentence is difficult to understand. Please revise
Line 73: Describing how patients are perceived by their environment - please change to "by their listeners" or "by their communicative partners"
Line 75: "say more about the ultimate treatment goals than about the ..." - this is also difficult to understand. Perhaps consider "point more to functional outcome goals than to specific therapeutic strategies...
Line 81: "they often have a significant share..." reads awkward. Consider changing to "they often contribute to ..."
Line 82: "to sustain a continuant sound" is an unusual expression. Consider changing this to "to sustain phonation for as long as possible"
Line 84: "a mouth movement at maximum rate" perhaps consider "fast repetitions of orofacial movements"
Line 171: "brain -damaged patients" - please use person first language "patients with brain damage"
Line 178: "spontaneously" - consider "on their own"
Line 179: "sustained vowel duration" - perhaps better "sustained phonation"
Methods
Line 357 "all examiners went through a training" - perhaps state here that the training will be described in more detail later on. I was wondering at this point what kind of training, but as I kept reading, I realized that it was explained.
I wondered why a 3 point scale was used for some perceptual ratings but a 5 point scale was used for others - what was the rationale for choosing a 3 point scale vs. a 5 point scale and why not a visual analog scale (as it is often used for intelligibility ratings/naturalness ratings)?
Why was "Dadada" but not "Gagaga" included in the study? I think the rationale was that tongue tip is representative for the tongue; however, based on clinical experience the tongue back movements are often more difficult. Thus, someone may argue that you may have found more effector-specificity if "gagaga" had been included? What do you think?
I loved how the statistical approach was explained. I was able to follow it although I was not super familiar with the statistical approach. Well done!
Figures are also very helpful.
Table 3. the 3 stars for the t-test results indicate a p-value but I cannot find what p-value threshold it indicates. Please add this information to the table header or subtitle of the table.
Figure 3. What is the x-axis ? please add an axis label.
Figure 4. Also needs an x-axis label. Perhaps consider scaling all three subplots the same on the x-axis for easier visual comparisons. That is, the NSOM subplot is on a different scale than the other two subplots.
Discussion
well organized and great content. I enjoyed reading it all.
Author Response
We thank the reviewer for the careful reading of our manucript and the valuable and constructive suggestions for revision. We have followed all suggestions as specified in the point-by-point responses below.
Rev. line 32: consider changing the sentence to "perceptual speech characteristics of dysarthria in patients with neurological disorders".
Au: Our study relates to perceptual and acoustic speech parameters, and the patient sample comprised individuals with movement disorders. We have therefore changed the reviewer’s suggested revision to “… to speech characteristics of dysarthria in individuals with movement disorders”
Rev. line 35: consider "auditory-perceptual ratings and speech acoustic measures"
Au: Thanks for the suggestion. We would prefer to retain our formulation, in order to keep "auditory perceptual" and "acoustic" as equal-ranking attributes of "speech."
Rev. line 37: consider replacing "gender" with "sex" throughout the paper because gender refers to the social construct while sex refers to the biological attribute.
Au: done
Rev. line 38: "were used " instead of "was used"
Au: thanks, - done
Rev. line 39: The sentence "in a first analysis we tested...." is really difficult to understand because it is so long. Perhaps try to split up into several sentences and rephrase.
Au: done
Rev. line 43: the expression "decompose along ..." is difficult to understand. Perhaps rephrase?
Au: done
Rev. line 45: "diagnostic measures of speech characteristics in PWD" - I have had trouble throughout the manuscript with the terms "measure", "parameter" and "variable" - they seem to be used interchangeably but it would help if the use of these words were more consistent. What is a parameter, what is a measure, and what is a variable?
Au: The term “parameter” was mainly used to describe established elements of speech assessments or diagnostic profiles (e.g., “nonspeech parameters currently used in dysarthria diagnostics”), whereas “measures” was used in more general contexts related to the measurement of some functional speech or nonspeech characteristics (e.g., “the parameter PAUS is a measure of fluency”). “Variables” denotes the placeholders in a statistical computation or a graphic representation of diagnostic parameters. We screened the manuscript for inconsistencies regarding these conventions and modified the terminology as appropriate.
Rev: line 51: Suggestion to change to "Dysarthria is a complex neurogenic motor speech disorder"
Au: done
Rev: line 54: As a result of these deficits... - this sentence is difficult to comprehend. Perhaps add "may sound less "natural", and may require more effort and attention from conversation partners"
Au: done
Rev: Line 67: As a signature of impaired speech motor functions, these parameters are essential... - the reference to "these parameters" is unclear and the sentence is difficult to understand. Please revise
Au: done
Rev: Line 73: Describing how patients are perceived by their environment - please change to "by their listeners" or "by their communicative partners"
Au: done (“… by their interlocutors”)
Rev: Line 75: "say more about the ultimate treatment goals than about the ..." - this is also difficult to understand. Perhaps consider "point more to functional outcome goals than to specific therapeutic strategies...
Au: done. Thanks for the suggestion.
Rev: Line 81: "they often have a significant share..." reads awkward. Consider changing to "they often contribute to ..."
Au: done
Rev: Line 82: "to sustain a continuant sound" is an unusual expression. Consider changing this to "to sustain phonation for as long as possible"
Au: done
Rev: Line 84: "a mouth movement at maximum rate" perhaps consider "fast repetitions of orofacial movements"
Au: done (with some adaptation to the syntactical structure of the whole passage)
Rev: Line 171: "brain -damaged patients" - please use person first language "patients with brain damage"
Au: done
Rev: Line 178: "spontaneously" - consider "on their own"
Au: done
Rev: Line 179: "sustained vowel duration" - perhaps better "sustained phonation"
Au: done
Rev: Line 357 "all examiners went through a training" - perhaps state here that the training will be described in more detail later on. I was wondering at this point what kind of training, but as I kept reading, I realized that it was explained.
Au: done
Rev: I wondered why a 3 point scale was used for some perceptual ratings but a 5 point scale was used for others - what was the rationale for choosing a 3 point scale vs. a 5 point scale and why not a visual analog scale (as it is often used for intelligibility ratings/naturalness ratings)?
Au: 3-point-scale ratings were applied to evaluate short responses, such as consonant accuracy in single word- or syllable productions (parameters WORD, SYLL) or isolated orofacial movements (parameter SNGL). A finer evaluation of such responses (e.g., by a 5-point or a VA scale) is not viable. In contrast, 5-point rating scales have proven useful in the BoDyS protocol, where they are applied to utterances that usually include more than one intonation phrase. We have decided not to add such methodological details to the already very extensive methods section.
Rev: Why was "Dadada" but not "Gagaga" included in the study? I think the rationale was that tongue tip is representative for the tongue; however, based on clinical experience the tongue back movements are often more difficult. Thus, someone may argue that you may have found more effector-specificity if "gagaga" had been included? What do you think?
Au: Thanks for raising this issue. Our syllable repetition protocol deviates from the standard protocol in several respects. We have now included footnote 1 to explain the rationale of using /ba/, /da/, and /bada/.
Rev: I loved how the statistical approach was explained. I was able to follow it although I was not super familiar with the statistical approach. Well done!
Figures are also very helpful.
Au: Thank you for the favorable judgment.
Rev: Table 3. the 3 stars for the t-test results indicate a p-value but I cannot find what p-value threshold it indicates. Please add this information to the table header or subtitle of the table.
Au: This comment refers to a table that should actually be numbered as Table 4. We have adjusted table numbers and have added the p-value in the subtitle of table 4. Sorry for this inaccuracy.
Rev: Figure 3. What is the x-axis ? please add an axis label.
Au: Done.
Rev: Figure 4. Also needs an x-axis label. Perhaps consider scaling all three subplots the same on the x-axis for easier visual comparisons. That is, the NSOM subplot is on a different scale than the other two subplots.
Au: Done.
Rev: Discussion: well organized and great content. I enjoyed reading it all.
Au: Thank you for all the time you have spent reviewing this paper and for your constructive and positive feedback
Reviewer 2 Report
The study presented in this paper investigates if and how “nonspeech” measures usually used in the clinical assessment of motor speech disorders relate to the characteristics of speech impairment in patients with dysarthria. The introduction nicely presents the two contrasting views about the relationship between speech and non-speech tasks: the dominant view that non-speech tasks are relevant to the assessment of motor speech impairment, that are thought to be domain-general and that non-speech tasks allow to a detailed assessment of some specific effector-specific motor impairments that can less easily been tease apart with speech tasks and the alternative view claiming that different motor control systems are involved in speech and non-speech movements even when the same effectors are involved. The authors thus provide clear argumentation in favor of the need for empirical evidence for the validity of the use of nonspeech tasks in assessment, in particular given their functional non usefulness and the lack of converging/coherent results so far, leading to clear formulation of the study goals.
Several non-speech measures were compared to perceptual and acoustic speech measures in 130 patients with neurological motor impairment with standard scores computed relative to a group of 130 neurotypical speakers. Exploratory graph analyses run on 23 speech and nonspeech measures show a separation between speech and nonspeech factors; the same results are observed on an additional analyses run on the same measures reorganized according to specific effectors involved in the tasks. Based on these results the authors conclude against the view that nonspeech parameters used in dysarthria assessment are congruent with diagnostic measures of speech and claim that nonspeech tasks should not be part of the standard clinical assessment of dysarthria.
The study clearly fills a gap in the understanding of the relationship between speech and nonspeech measures in dysarthria and has important clinical impacts. I only have a few issues, that however have an impact on the strength of the claims that may be moderated at some points.
A/In the conclusion the authors claim that “in the interest of diagnostic economy (…), we suggest that these tasks be used only when there is credible evidence about how they contribute to clinical decision making (…)”-- > A limit of the study relative to this conclusion is that patients from 6 different underlying neuropathology were pulled together and one cannot exclude that for some subgroups there may be a relationship between impaired speech and nonspeech factors. If an analysis by subgroups cannot be run due to small samples per subgroup, this issue should at least be acknowledged and discussed.
B/The interpretation of some results is also unclear or may have been disregarded:
i. In the CFA confirmation analysis the correlation between the dimensions in the first EGA analyses indicate quite strong correlation between SPEECH2 and DDK. This is the strongest correlation, meaning that it is larger than the correlation between the three SPEECH dimensions. It is unclear how this result is interpreted by the authors relative to the weak loadings between speech and DDK in the EGA analysis.
ii. Only 79 out of 130 patients are below the BoDyS cut-off for dysarthria, meaning that analyses include 23 x 51 measures from “typical speakers” (i.e. many standard scores around 0 ) and 23 x 79 measures from dysarthria,: are the results replicated when the non-dysarthric participants are removed from the analyses?
Minor issues
INTRODUCTION
- In the section 1.1.2. an additional argument that may be mentioned are dissociations and double dissociations between impairments in non-speech and in speech tasks.
METHOD
- “Each parameter was standardized for its variation in the control group and for possible age- and gender-influences.” : what is the age factor and how many control participants per group if it is a group factor? Please clarify how the standardization relative to age was dealt.
- The same question holds for results presented in Table 3 and Fig 1: is every patient z-score meant against the reference population in age and gender?
- “A control group of 130 healthy subjects (67 women, median age 49, range 18 to 90; 342 evenly distributed over decades” à related to a previous comment: clarify, as age is one of the standardization parameters
- section 2.2. Assessments and evaluation à this section is a bit hard to read/understand at it refers to materials that are presented only later, suggest to inverse and it is unclear which tasks measures are taken from BoDyS and which are additional tasks administered to the participants. For instance “Elicitation of text-, sentence- and word production was conducted in an audio-sup-358 ported reading format”: which tasks, which text, sentences and words?
- Same issue with section 2.2.2. Articulation rate (RATE): “reading task described above”, the reading task is not described above.
- DDK syllables and lip/tongue : for how long were participants asked to repeat the syllable/movement?
- DDK syllables: Although ba, da and bada have been previously used by the same authors, the reader might be reminded of why those syllables are favored over the most commonly used (i.e. pa, ta, ka and pataka)
- DDK lip/tongue: I wonder if calling this task “diadochokinetic lip- and tongue movement rate” is appropriate. The DDK task is characterized by the repetition at a maximum rate and in a single breath. In the diadochokinetic lip- and tongue movement task you mention that there is no particular instruction concerning the respiratory activity, which is why the task might not meet the criteria of a DDK task.
-
RESULTS
- In the CFA analyses, the correlation between dimensions are mentioned only for the highest and lowest correlation, could the authors provide all the correlations in the appendix (see also main comment B.i.
OTHER
Line 674: to the five dimensionà dimensions
Author Response
We thank the reviewer for the careful reading of our manucript and the valuable and constructive suggestions for revision. We have followed all suggestions as specified in the point-by-point responses below.
Rev: … The study clearly fills a gap in the understanding of the relationship between speech and nonspeech measures in dysarthria and has important clinical impacts. I only have a few issues, that however have an impact on the strength of the claims that may be moderated at some points.
Au: Thanks for your favorable evaluation
Rev: A/In the conclusion the authors claim that “in the interest of diagnostic economy (…), we suggest that these tasks be used only when there is credible evidence about how they contribute to clinical decision making (…)”-- > A limit of the study relative to this conclusion is that patients from 6 different underlying neuropathology were pulled together and one cannot exclude that for some subgroups there may be a relationship between impaired speech and nonspeech factors. If an analysis by subgroups cannot be run due to small samples per subgroup, this issue should at least be acknowledged and discussed.
Au: Thanks for raising this point. Indeed, the relationship between speech and certain nonspeech parameters varies between etiologies (as has already been shown elsewhere), which is incompatible with the assumption of domain-general core functions underlying speech and nonspeech tasks. The point is briefly mentioned in section 1.3, but not elaborated any further in the article so far. We are currently preparing a follow-up paper that addresses exactly this issue (as well as two other issue brought up in your comments, - see below). To consider these questions adequately would have exceeded the scope of the present paper. We have now added this point as an open issue at the end of section 4.
Rev: B/The interpretation of some results is also unclear or may have been disregarded:
- In the CFA confirmation analysis the correlation between the dimensions in the first EGA analyses indicate quite strong correlation between SPEECH2 and DDK. This is the strongest correlation, meaning that it is larger than the correlation between the three SPEECH dimensions. It is unclear how this result is interpreted by the authors relative to the weak loadings between speech and DDK in the EGA analysis.
Au: This is a further interesting point that we are going to address in the follow-up paper mentioned above. As reported in section 1.3, DDK rate and articulation rate are uncorrelated in healthy adult speakers, – for obvious reasons, because a person’s habitual speaking rate does not necessarily predict how fast they are able to repeat PA or KA in DDK testing. If the two measures are found to be correlated in patients with movement disorders, this may say more about the patients’ maximum performance abilities than about speech. We have added a comment at the end of section 4.
Rev: ii. Only 79 out of 130 patients are below the BoDyS cut-off for dysarthria, meaning that analyses include 23 x 51 measures from “typical speakers” (i.e. many standard scores around 0 ) and 23 x 79 measures from dysarthria,: are the results replicated when the non-dysarthric participants are removed from the analyses?
Au: A unique feature of this study is that it included patients in whom dysarthria may be suspected but who did not necessarily have dysarthria. In this sense, the study simulates the situation of a clinician who is not sure whether the patient to be diagnosed is dysarthric or not. Many earlier approaches have dealt with patients with a given clinical diagnosis of dysarthria, thereby circumventing the problem that there is no universally accepted criterion to decide whether a person has (mild) dysarthria or still falls within the (lower) range of "normal" or "typical" speech. The data reported in the present paper may be considered to represent the distributions of diagnostic scores that can occur in clinical assessments of neurologic patients, while the question of how many individuals in the PAT group actually had dysarthria is of minor relevance.
The concern that our data sample is predominated by scores around 0 is actually unwarranted, as the strongly left-skewed distributions in Fig. 1 illustrate. We have now also included histograms of the BoDyS standard scores of the CTRL and PAT groups (Figure C-2 in the Appendix) to illustrate that only a small proportion of the patients had standard scores around 0 (or > 0), indicating that most of the patients who scored within the CTRL range still had relatively low BoDyS scores. What this may mean is beyond the scope of the present paper and will be the subject of the follow-up article already mentioned. The cited number of 79/130 most likely having dysarthria was based on a very conservative 1%-cutoff criterion and is not doing justice to the obvious diagnostic uncertainty. We have therefore expanded on this point in section 3.1 by elucidating further details of the distribution of BoDyS scores, especially the prevalence rate of 70% when a 5% cutoff is applied. However, we do not want to be more precise in this article about which of the participants were dysarthric and which were not.
Rev: In the section 1.1.2. an additional argument that may be mentioned are dissociations and double dissociations between impairments in non-speech and in speech tasks.
Au: Thanks for reminding us that we forgot to mention this important argument. We have now included a short paragraph at the beginning of section 1.3, where it fits better than in 1.1.2, in our opinion
Rev: “Each parameter was standardized for its variation in the control group and for possible age- and gender-influences.” : what is the age factor and how many control participants per group if it is a group factor? Please clarify how the standardization relative to age was dealt.
Au: As said in the paragraph following this sentence, standardization was performed by regression. We have now added the information that age was included as a continuous factor. The distribution of age in the CTRL group is plotted in Figure C-1 now in the Appendix.
Rev: The same question holds for results presented in Table 3 and Fig 1: is every patient z-score meant against the reference population in age and gender?
Au: We now added the information that all variables were corrected for variations by age and sex at the beginning of section 3.1.
Rev: “A control group of 130 healthy subjects (67 women, median age 49, range 18 to 90; 342 evenly distributed over decades” à related to a previous comment: clarify, as age is one of the standardization parameters
Au: Age was evenly distributed between 18 and 90 years. We have now added Figure C-1 in the Appendix and an explanation in the text.
Rev: section 2.2. Assessments and evaluation à this section is a bit hard to read/understand at it refers to materials that are presented only later, suggest to inverse and it is unclear which tasks measures are taken from BoDyS and which are additional tasks administered to the participants. For instance “Elicitation of text-, sentence- and word production was conducted in an audio-sup-358 ported reading format”: which tasks, which text, sentences and words?
Au: We have clarified this by summarizing the materials used in speech examinations beforehand.
Rev: Same issue with section 2.2.2. Articulation rate (RATE): “reading task described above”, the reading task is not described above.
Au: Clarified by referring to the materials used in the BoDyS examination.
Rev: DDK syllables and lip/tongue : for how long were participants asked to repeat the syllable/movement?
Au: The syllable DDK was interrupted when the patient had to take a breath (“… on a single breath”; line 483; “… we analyzed a maximum of ten and a minimum of six syllables, uninterrupted by pauses”, line 485). For the lip- and tongue movements, “a maximum of four and a minimum of three cycles per task was analysed” (line 506). We have now added a short explanation.
Rev: DDK syllables: Although ba, da and bada have been previously used by the same authors, the reader might be reminded of why those syllables are favored over the most commonly used (i.e. pa, ta, ka and pataka)
Au: A similar point was also raised by Rev. #1. We have now included Footnote 1 to explain why we used unaspirated plosives and why we avoided 3-syllabic syllable repetitions (pataka).
Rev: DDK lip/tongue: I wonder if calling this task “diadochokinetic lip- and tongue movement rate” is appropriate. The DDK task is characterized by the repetition at a maximum rate and in a single breath. In the diadochokinetic lip- and tongue movement task you mention that there is no particular instruction concerning the respiratory activity, which is why the task might not meet the criteria of a DDK task.
Au: In this terminology we followed the review paper by Kent et al. (Ref. [39]). The term “DDK” has originally been used in neurologic examinations of the upper extremities.
Rev: In the CFA analyses, the correlation between dimensions are mentioned only for the highest and lowest correlation, could the authors provide all the correlations in the appendix (see also main comment B.i.
Au: We have added correlation matrices in Appendix D and a comment at the end of section 4 (see above).
Rev: Line 674: to the five dimensionà dimensions
Au: Corrected. Thank you.
Thank you for all the time you have spent reviewing this paper and for your constructive and positive feedback.